# Assessing the Utility of Neonatal Screening Assessments in Early Diagnosis of Cerebral Palsy in Preterm Infants [note 1]

**DOI:** 10.3390/brainsci12070847

**Published:** 2022-06-28

**Authors:** Rebecca Connors, Vathana Sackett, Catherine Machipisa, Kenneth Tan, Pramod Pharande, Lindsay Zhou, Atul Malhotra

**Affiliations:** 1Department of Paediatrics, Monash University, Melbourne, VIC 3800, Australia; rcon0002@student.monash.edu (R.C.); kenneth.tan@monashhealth.org (K.T.); lindsay.zhou@monashhealth.org (L.Z.); 2Allied Health Department, Monash Children’s Hospital, Melbourne, VIC 3168, Australia; vathana.sackett@monashhealth.org (V.S.); catherine.machipisa@monashhealth.org (C.M.); 3Monash Newborn, Monash Children’s Hospital, Melbourne, VIC 3168, Australia; pramod.pharande@monashhealth.org; 4Early Neurodevelopment Clinic, Monash Children’s Hospital, 246 Clayton Road, Melbourne, VIC 3168, Australia

**Keywords:** CP, GMA, HINE, HINE, prediction, therapies

## Abstract

Background: Early diagnosis of cerebral palsy (CP) in high-risk infants is possible at 3–4 months’ corrected age (CA) using standardised assessments. Aim: To assess the utility of neonatal screening assessments—writhing general movements (GMs) and the Hammersmith Neonatal Neurological Examination (HNNE)—to predict CP/high-risk status at 3–4 months’ CA in extremely preterm infants. Methods: Retrospective cohort study of high-risk preterm infants (born < 29 weeks’ gestation and/or birth weight < 1000 g) attending an Early Neurodevelopment Clinic. Data from neonatal assessments were compared with CP/high-risk diagnosis at 3–4 months’ CA, fidgety GMs, and Hammersmith Infant Neurological Examinations (HINE) using logistic regression, linear regression, and Spearman rank correlation. Results: Two hundred and two preterm infants (median gestation age at birth 27.3 (IQR 25.4–28.3) weeks, mean birth weight 870.3 (SD 248.4) grams) were included. A total of 26 (12.8%) infants received early CP/high-risk diagnoses at 3–4 months’ CA. A lower gestational age (GA) (OR = 0.78; *p* = 0.029, 95% CI [0.26, 0.97]) and abnormal writhing GMs (OR 1.56; *p* = 0.019, 95% CI [1.07, 2.27]) were predictive of CP/high-risk diagnosis. Although after adjustment for sex, GA, birth weight, and growth restriction, GA (aOR = 0.67; *p* = 0.068, 95% CI [0.44, 1.03]) and writhing GMs (aOR = 1.44; *p* = 0.087, 95% CI [0.95, 2.20]) were not significant, a strong trend still persisted. The HNNE scores significantly correlated with both the HINE evaluation (r_s_ = 0.43, *p* < 0.001, 95% CI [0.31, 0.56]) and fidgety GMs (r_s_ = −0.10, *p* = 0.012, 95% CI [−0.32, −0.04]). Linear regression confirmed the HNNE was highly predictive of the HINE (correlation coefficient 0.82; *p* < 0.001, 95% CI [0.48, 1.15]). Writhing GMs did not significantly correlate with either fidgety GMs (*p* = 0.723, 95% CI [−0.12, 0.17]) or the HINE (*p* = 0.173, 95% CI [−0.24, 0.04]). Conclusions: Abnormal writhing GMs in the neonatal period were non-significantly associated with early CP/high-risk diagnoses in extremely preterm infants in a multivariate analysis. Additionally, the HNNE significantly correlated with both fidgety GMs and the HINE.

## 1. Introduction

Cerebral palsy (CP) is the most common physical disability diagnosed in children, with an overall prevalence of approximately 2.0–2.11/1000 live births internationally and 1.4/1000 live births in Australia [1,2]. Preterm birth is a known risk factor for CP, with a prevalence rate of 62.5/1000 live births of infants born between 22 and 31 weeks’ gestation, compared to 0.9/1000 live births for infants born over 37 weeks’ gestational age (GA) [3]. 

The early diagnosis of CP is now possible in high-risk infants at 3–4 months’ corrected age (CA). Early intervention may promote positive cognitive and motor developmental outcomes amongst high-risk infants, and thus the early diagnosis or early identification of infants at high risk of CP is a critical step towards enabling this [4,5].

There is a range of assessment tools currently utilised to detect movement difficulties and predict CP in infants. Prechtl’s general movement assessments (GMA), Hammersmith Infant Neurological Examinations (HINE), and MRI brain scans have the highest predictive validity before 5 months’ corrected age and are recommended to be used to predict CP with high sensitivity before 5 months’ corrected age [6,7,8]. Notably, fewer than 5% of reported cases of CP are false positives, and the majority of these cases resulted in re-diagnosis with another neurological deficit, rather than a normal neurological outcome [7]. 

The Prechtl GMA is an observational tool used to assess the spontaneous movements of an infant, known as general movements (GMs). GMs arise during early foetal life and persist until 4–5 months’ CA. GMs can be recorded after a preterm birth until 6 weeks’ CA. Known as “writhing movements”, these are characterised by variable movements through the whole body, including arm, leg, neck, and trunk movements. Normal writhing movements appear smooth and unpredictable and show variability in the intensity, range, direction, and complexity of movement [9,10]. Abnormal writhing movements may be classed as poor-repertoire (PR), cramped-synchronised (CS) or chaotic (Ch), and are characterised by a lack of fluency, complexity, variability, and frequency of movements [9]. Between 10 weeks and 4–5 months’ CA [11], GMs are known as fidgety movements (FMs) and are classified as normal, abnormal, sporadic, or absent, with absent FMs being the most strongly correlated with abnormal outcomes. The most accurate time for the assessment of GMs during this fidgety period is at 12–14 weeks’ CA [12]. Several systematic reviews [6,7,13,14] have validated GMs as a reliable predictor of CP, although this primarily relies on assessments performed during the fidgety period, due to reported high false-positive rates of writhing GMs [6,7,13,14,15,16]. 

The Hammersmith Infant Neurological Examination (HINE) is a validated scored assessment of 26 items, each scored from 0 to 3, used to evaluate infants between 2 and 24 months’ CA to inform a diagnosis of CP. It functions as both a quantitative and qualitative assessment, evaluating movement, cranial nerves, tone, reflexes, and posture [17]. A global cut-off score of <57 out of a maximum of 78 at 3 months’ CA is widely accepted as predictive of CP at 3–4 months’ CA with high sensitivity (90–96%) and specificity (85–87%) [7,8,17,18].

Currently, a combination of GMA, the HINE, and neonatal MRI is the gold standard for the diagnosis of CP in high-risk infants and can be used to accurately predict CP before 5 months’ CA [6,7,8,14].

The Hammersmith Neonatal Neurological Examination (HNNE) is a scored neurological assessment presently limited to use as a screening tool in term and preterm infants, though it has not been validated for use in infants <34 weeks’ GA. It consists of 34 items across six domains: tone (10 items), tone patterns (5 items), reflexes (6 items), spontaneous movements (3 items), abnormal signs (3 items), and behaviour (7 items), with a maximum global optimality score of 34 [19,20]. In an analysis of low-risk term infants, Dubowitz et al. concluded that an optimality score of <31 identifies infants requiring a follow-up assessment but is not conclusive of neurological abnormality [19]. Since then, the examination has been validated for use in preterm infants at term-equivalent age with an optimality score of 27 [20,21,22]. Data on the reliability of the HNNE to assess extremely preterm infants in the preterm period (<37 weeks’ GA) are lacking [20]. 

Several studies have identified a relationship between GMs and CP diagnosis amongst very preterm infants assessed at preterm or term-equivalent age [15,23,24]. However, there are limited data analysing the utility of early neonatal assessments performed in the extremely preterm period to support the accurate identification of preterm infants at high risk of CP during this period. Given the established importance of early intervention to improve outcomes in CP and the increased risk of developing CP amongst more preterm infants, there is a need to validate reliable indicators of CP among high-risk infants in the preterm period to avoid delaying intervention [7,24,25]. Additionally, in some cases, inter-hospital transfer, travel requirements, or clinic availability may preclude infants from receiving a follow-up at 3–4 months’ CA, and therefore, neonatal assessments offer an important opportunity to identify high-risk infants that may otherwise be missed.

Presently, no studies have analysed the utility of early neonatal assessments in the preterm period for predicting the later diagnoses made based on fidgety age assessments in very preterm (<29 weeks) or extremely low birth weight (1000 g) infants at 3–4 months of corrected age. There are also no studies tracking the correlation between the HNNE and HINE scores.

We conducted an exploratory study to assess how early neonatal screening assessments correlate with 3–4-months’ CA assessments of high-risk infants in the setting of a clinical surveillance clinic (Early Neurodevelopment Clinic (ENC)), in order to answer the question: can early neonatal screening assessments accurately predict early diagnoses of CP/high-risk status in high-risk infants at 3–4 months’ corrected age in a real-world setting? The answer to this question may be useful in guiding the approach to high-risk infants and maximising early intervention at term age to improve outcomes amongst the high-risk preterm cohort, or in guiding further research that needs to be undertaken. 

## 2. Methods

### 2.1. Study Design

This retrospective cohort study analysed the data of high-risk preterm infants (born < 29 weeks’ gestation and/or birth weight < 1000 g) attending the Early Neurodevelopment Clinic at Monash Children’s Hospital, Melbourne, Australia. This clinic was established in 2018 to facilitate the early diagnosis of CP in line with current recommendations [7].

Data were collected from electronic records of clinic notes between May 2019 and November 2021. The records of 393 patients attending ENC were reviewed; 190 did not fit the inclusion criteria (<29 weeks/<1000 g) and were excluded. One further patient was excluded due to non-attendance at their ENC appointment. Early neonatal assessment results and demographic information were obtained from the electronic patient records of the remaining 202 infants who attended the clinic. Assessments were performed and videos were scored in practice by experienced clinicians at Monash Newborn and ENC, and videos were not re-reviewed during the study. Where practicable, infants had multiple writhing GM videos captured and assessed during their neonatal admission between 32 and 46 weeks’ GA. Thirty infants had only one writhing GM video recorded during their admission. Factors preventing multiple recordings included short admission duration, inter-hospital transfer, and patient behaviour unsuitable for assessment. The HNNE was performed between 35 and 45 weeks’ GA. 

Participants were divided into two cohorts based on their diagnosis at 3–4 months’ CA:CP or high-risk for developing CP, and normal development or developmental delay other than CP. Diagnoses were made based on fidgety GMA and the HINE in combination with neuroimaging findings, such as cranial ultrasounds or MRI brain scans, where available. CP and high-risk status were distinguished as some infants meeting criteria for CP (absent fidgety GMs, HINE < 57) did not receive a definitive diagnosis at 3–4 months’ CA but were still referred for close follow-up to monitor progression. Neonatal neuroimaging findings were included in the 3–4-month assessment for CP, and thus cannot be included in the comparison between these two time points. Therefore, this paper focuses on clinical assessments.

All available writhing GM scores from the neonatal period were reviewed, and patients were categorised according to the worst recorded score, as follows: Normal = 0;PR = 1;PR/CS = 2;CS on one occasion = 3;CS on two or more occasions = 4.

The group for which CS > 2 was distinguished from the CS group to illustrate the difference between a one-off CS GM and a trajectory of persistent CS GMs. Of the 27 infants who received a score of 3, only 1 did not have multiple writhing GM assessments performed. 

Fidgety GMs were categorised as follows:Present = 0;Sporadic = 1;Abnormal = 2;Absent = 3.

Global cut-off scores of 27 and 57 were used for the HNNE and the HINE, respectively.

This study was exempt from review by Monash Health Human Research Ethics Committee and received ethics approval by Monash Health as a quality assurance project (ERM Reference No. 81908). It followed NHMRC Ethical Considerations in Quality Assurance and Evaluation Activities (2014) guidelines.

### 2.2. Statistical Analysis 

Statistical analyses were performed using Stata Version 16 (StataCorp LLC, College Station, TX, USA). Additional analyses were performed in GraphPad Prism Version 9.0 (GraphPad Software, San Diego, CA, USA). The Wilk–Shapiro test was performed to determine the normality of numerical data. The mean and standard deviation were calculated for normally distributed data, while the median and inter-quartile range were calculated for non-normally distributed data. 

Data from neonatal assessments (writhing GMs and HNNE scores) were correlated against results of 3–4-months’ CA assessments (fidgety GMs and HINE scores) using Spearman’s Rho correlation. Linear regression was performed to directly compare the HNNE with subsequent HINE scores. Neonatal assessment scores (writhing GMs and HNNE) were studied for association with the 3–4-months’ CA diagnosis using a parsimonious model univariate logistic regression and then adjusted for demographic characteristics (sex, gestational age, birth weight, and small for gestational age) with the results expressed as odds ratios, with 95% confidence intervals and significance set at <0.05. 

## 3. Results

### 3.1. Demographics

Two hundred and two preterm infants were included in the study. The median gestational age of participants was 27 + 3 weeks (IQR 25 + 4–28 + 3), and the mean birth weight was 870.3 g (SD 248.4 g). Twenty-six infants (12.8%) received a CP or high-risk clinic diagnosis—eight (4.0%) infants were diagnosed with CP, and eighteen (8.9%) were considered at high-risk. Eighty-two (40.6%) infants were deemed developmentally normal at ENC, and ninety-four (46.5%) were deemed delayed in specific parts of development but not high-risk for CP. Additional demographic characteristics of the participants are outlined in Table 1.

One infant did not receive a diagnosis at ENC. Eight did not receive HINE scores, and three did not receive fidgety GM scores. Thirty-three infants did not have available HNNE scores, and three did not have available writhing GMs. All infants attending the clinic were within the fidgety period (between 10 and 20 weeks’ CA) [11].

### 3.2. Primary Outcome—Univariate Analysis—Early Neonatal Assessments vs. CP/High Risk Diagnosis 

Parsimonious model univariate logistic regression (Table 2) determined that a lower gestational age (OR = 0.78; *p* = 0.029, 95% CI [0.26, 0.97]) and all abnormal writhing GMs (OR 1.56; *p* = 0.019, 95% CI [1.07, 2.27]) were significantly predictive of CP/high-risk diagnosis at 3–4 months’ CA. The relationship between writhing GMs and 3–4-month diagnosis is represented in Figure 1. Figure 2 shows the distribution of the HNNE scores (Figure 2a), gestational age (Figure 2b), and birth weight (Figure 2c) according to 3–4-month diagnosis.

This trend persisted in multivariate analyses (Table 3), though the results were not statistically significant for GA (aOR = 0.67; *p* = 0.068, 95% CI [0.44, 1.03]) or writhing GMs (aOR = 1.44; *p* = 0.087, 95% CI [0.95, 2.20]).

### 3.3. Secondary Outcome—Early Neonatal vs. Infant Assessments 

Correlation analysis (Table 4) showed that the HNNE had a Spearman’s coefficient (r_s_) of 0.43 for the HINE (*p* < 0.001, 95% CI [0.31, 0.56]) and −0.18 for fidgety GMs (*p* = 0.012, 95% CI [−0.32, −0.04]), indicating a significant correlation between the HNNE and both the HINE and fidgety GM scores. Linear regression (Table 5; Figure 3) confirmed the relationship between the HNNE and HINE, with a correlation coefficient of 0.82 (*p* < 0.001, 95% CI [0.48, 1.15]).

The writhing GMs had a non-significant Spearman’s Rho coefficient of 0.03 for fidgety GMs (*p* = 0.723, 95% CI [−0.12, 0.17]) and −0.10 for the HINE (*p* = 0.173, 95% CI [−0.24, 0.04]). 

Note that the GM scales used to categorise participants for both fidgety and writhing GMs increased with worsening performance, hence the negative coefficients. 

## 4. Discussion

The early diagnosis of CP in high-risk infants is a key priority in order to facilitate early intervention and maximise patient outcomes. The findings of this study provided useful data on the utility of early neonatal assessments in predicting the early diagnoses of CP. The current literature available on the topic is based on diagnosis at two years of age or beyond. This study is unique in specifically assessing the utility of early neonatal assessments for predicting the 3–4-months’ CA early diagnosis. Our study found a non-significant association between abnormal writhing GMs and CP/high-risk diagnosis at 3–4 months’ CA when adjustments for demographic factors were included. The interim univariate analysis before adjustment showed a significant association between these two results, which indicates a trend. Other salient findings are the significance of reduced gestational age in predicting 3–4-month CP diagnosis, the significance of which was also reduced with multivariate analysis, as well as a strong correlation between the HNNE and both the HINE and fidgety GM scores. 

A review of the literature found fidgety GMs had sensitivity between 95 and 100% with a specificity of 89–98% for predicting CP diagnosis at 2 years of age [6,13,14,26]. Writhing GMs had a sensitivity range between 93 and 100% [13,14,16], but low specificity of 40–59% [14,16] and a positive predictive value (PPV) ranging between 8 and 68% [14,16], indicating a high proportion of false positives, as abnormal early GMs can normalise over time, though low false negatives with a negative predictive value of 80–100% [14,16]. In particular, cramped-synchronised writhing movements have a strong association with CP development, with a high positive likelihood ratio of 23.43 [14] and an odds ratio of 323.00 and 178.20 at preterm and term age, respectively [27]. However, there is some inconsistency in the data available on cramped-synchronised movements, with a low sensitivity of 70% [14], suggesting a significant false-negative rate, but a high NPV of 74–94% [6.14], which suggests low false negatives. Our results are consistent with these data, as abnormal writhing GMs (defined as any GM score other than ‘normal’) were found to have an odds ratio of 1.56 for the early diagnosis of CP at 3–4 months’ CA (*p* = 0.019, 95% CI [1.07, 2.27]). However, once the data were adjusted for birth weight, sex, GA, and SGA/IUGR, this correlation was not statistically significant (aOR 1.44, *p* = 0.087, 95% CI [0.95, 2.20]). Abnormal writhing GMs were also found to be weakly correlated with abnormal fidgety GMs (including absent and abnormal scores) (r_s_ = 0.03, *p* = 0.723, 95% CI [−0.12, 0.17]) and abnormal HINE scores (r_s_ = −0.10, *p* = 0.173, 95% CI [−0.24, 0.04]). Both writhing and fidgety GMs were scored with normal being 0 and increasing scores indicating worsening performance, which is reflected in the negative coefficients with both the HINE and HNNE. 

Our study also found the HNNE scores to be somewhat predictive of early CP diagnosis at 3–4 months (OR= 0.88). Because the HNNE scores increase with improving performance, an odds ratio <1 indicates that an increasing HNNE score is associated with a decreased likelihood of early CP diagnosis. However, this association was not statistically significant (*p* = 0.05, 95% CI [0.78, 1.00]). This is consistent with current evidence, which suggests that the HNNE performed at two weeks post-term in term infants, or at term-corrected age in preterm infants, is highly predictive of a neurodevelopmental deficit [28]. A recent study by Venkata et al. found that the HNNE had an equivalent predictive value for predicting CP at 1 year when performed early (i.e., before discharge in preterm infants) and when performed at the recommended age as outlined above [28]. The HNNE had a sensitivity of 50–64% and specificity of 73–77% with a global optimality score cut-off of 32.5, as per Dubowitz [19,28]. PPV was 32–33% [28]. These results indicate some utility for the HNNE in predicting neurodevelopmental outcomes, including CP, prior to term-equivalent age [28].

Our study also identified gestational age at birth as a significant predictive factor in the development of CP. Reduced gestational age had an odds ratio of 0.78 (*p*= 0.029, 95% CI [0.62, 0.97]) for early CP diagnosis at ENC. As with the HNNE, the risk of CP decreases with increasing gestational age. It is known that reduced gestational age at birth is a risk factor for CP [29,30]; however, as all participants in our study were <29 weeks’ GA at birth, this validates the other results as indicators of risk independent of prematurity.

## 5. Limitations

The main limitations of this study are related to the retrospective study design and data collection. Some participants did not have neonatal assessments performed during their hospital admission. This is often related to difficulties with timing assessments around infants’ feeding and sleeping schedules, infant temperament, or discharge to other (step-down) hospitals. Three patients (1.5%) did not have scores for writhing GMs; thirty-three (16.3%) did not have scores for the HNNE. A small number of infants attending the ENC had incomplete 3–4-month assessments; three (1.5%) did not have recorded results for fidgety GMs, and eight (4.0%) did not have HINE scores. The reasons cited included telehealth appointments limiting the ability to perform assessments, infant irritability impeding the assessment, and parental reluctance. 

Additionally, whilst all participants attending ENC appointments were within the 10–20-week fidgety period, there was some variation in the age at the time of appointment, which may have impacted the data. This is due in part to the COVID-19 pandemic, which resulted in some patient clinic appointments needing to be rescheduled due to lockdown requirements or moved to telehealth, which also impacted the examination accuracy. However, it is reflective of the real-world setting in which these assessments are conducted, and thus still provides valuable insight into their utility.

## 6. Conclusions

The findings of this study suggest that early neonatal assessments, in particular writhing GMs, are non-significantly associated with the early diagnosis of CP at 3–4 months’ CA. Writhing GMs and the HNNE may have combined value as supportive assessments for infants at risk of poorer outcomes at 3–4 months. This information may be utilised to identify infants who are likely to be diagnosed with CP or high-risk status from a high-risk population and ensure that long-term follow-up is planned from inpatient discharge. Additionally, it may be beneficial in organising access to early intervention and support services for high-risk infants.

## Figures and Tables

**Figure 1 brainsci-12-00847-f001:**
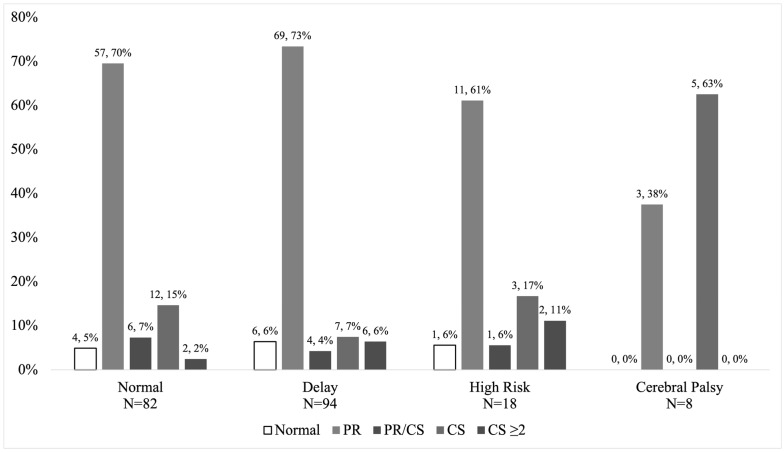
Distribution of writhing GM scores categorised by 3–4 months’ CA diagnosis.

**Figure 2 brainsci-12-00847-f002:**
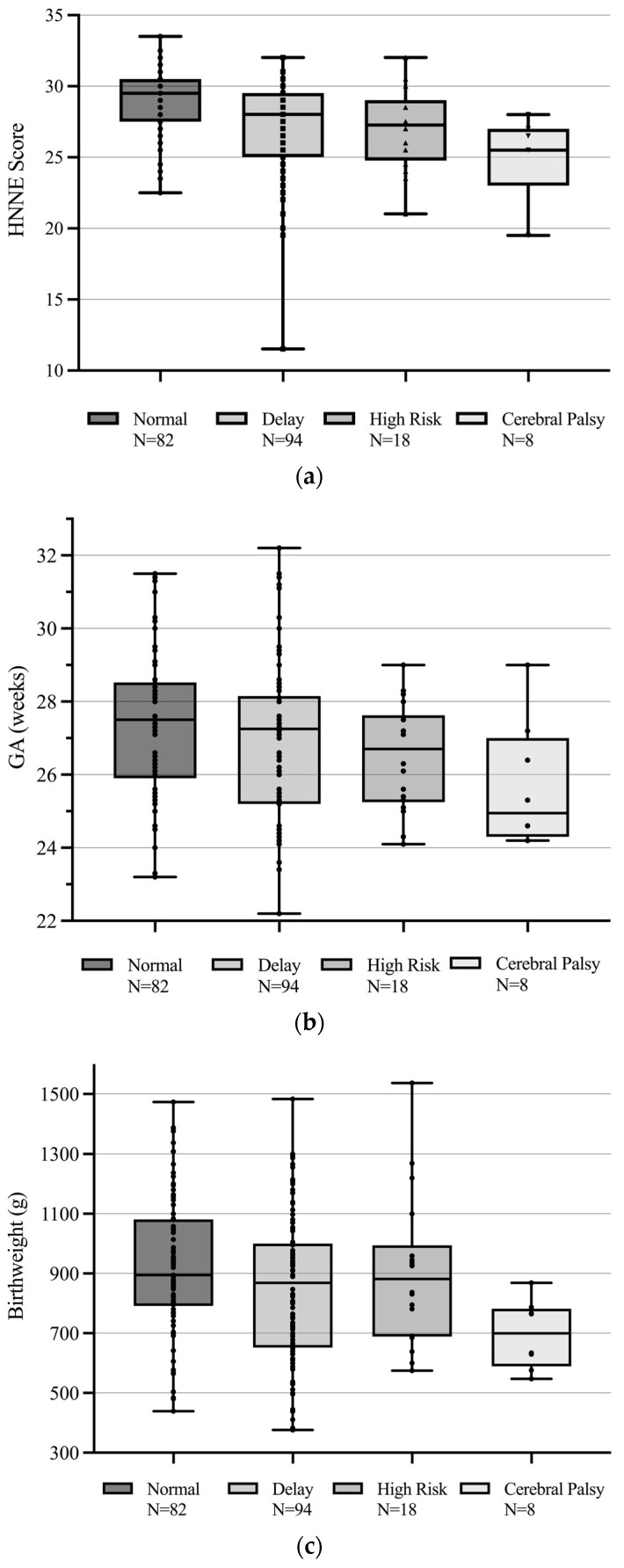
(**a**–**c**) Distribution of (**a**) HNNE scores, (**b**) gestational age, and (**c**) birth weight according to 3–4 months’ CA diagnosis.

**Figure 3 brainsci-12-00847-f003:**
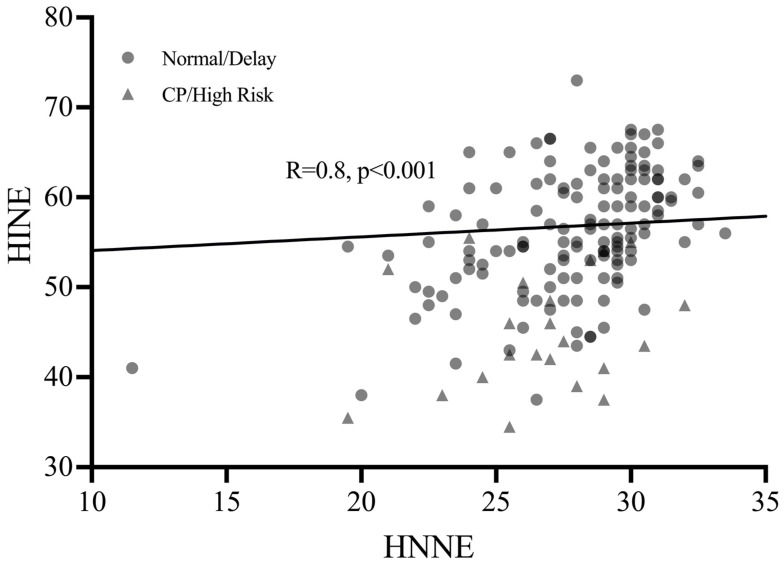
Linear Regression between HNNE and HINE at 3–4 months’ CA.

**Table 1 brainsci-12-00847-t001:** Patient characteristics.

Patient Characteristics	No Cerebral Palsy Diagnosis	Cerebral Palsy or High-Risk Diagnosis	*p*-Value
	N = 176	N = 26	
**Sex, *n* (%)**			
Male	83 (47.2)	15 (57.7)	0.32
Female	93 (52.8)	11 (42.3)	
**Gestational Age (w + d), median (*IQR*)**			
	27 + 4 (25 + 5, 28 + 4)	26 + 2 (26, 27 + 2)	0.018
**Birth weight (g), mean (*SD*)**			
	883.63 (242.03)	839.62 (237.44)	0.39
**SGA/IUGR, *n* (%)**			
	62 (35.2)	8 (30.8)	0.66
**Writhing GMA, *n* (%)**			
Normal	10 (5.8)	1 (3.8)	0.078
PR	126 (72.8)	14 (53.8)	
PR/CS	10 (5.8)	1 (3.8)	
CS	19 (11.0)	8 (30.8)	
CS ≥ 2	8 (4.6)	2 (7.7)	
**HNNE score, median (IQR)**			
	N = 147	N = 23	
	28.5 (26.5, 30)	27 (24.5, 29)	0.018
**Fidgety GMA, *n* (%)**			
Fidgety Present	144 (82.8)	2 (8.0)	
Fidgety Present (sporadic)	20 (11.5)	4 (16.0)	
Abnormal Fidgety	2 (1.1)	0 (0.0)	
Fidgety Absent	8 (4.6)	19 (76.0)	
**HINE score, median (IQR)**			
	N = 168	N = 26	
	57 (52.5, 62)	43 (39, 48.5)	
**Diagnosis at 3–4 months’ CA, *n* (%)**			
Normal	82 (46.6)	0 (0.0)	
Delay	94 (53.4)	0 (0.0)	
High Risk	0 (0.0)	18 (69.2)	
CP	0 (0.0)	8 (30.8)	

Data represented as number (%), median (IQR) or mean (SD). w + d, week/s + day/s; IQR, interquartile range; SD, standard deviation; SGA, small for gestational age; IUGR, intrauterine growth restriction; GMA, general movement assessment; PR, poor repertoire; CS, cramped-synchronised; HNNE, Hammersmith Neonatal Neurological Examination; HINE, Hammersmith Infant Neurological Exam; CA, corrected age; CP, cerebral palsy.

**Table 2 brainsci-12-00847-t002:** Parsimonious model univariate logistic regression between early neonatal assessments/demographics and early diagnosis of CP/high-risk status at 3–4 months’ CA.

	N	OR	Std. Err.	*p*-Value	95% CI
**Sex**	202	0.65	0.28	0.318	0.28–1.50
**GA**	202	0.78	0.09	0.029	0.62–0.97
**BW**	202	1.00	0.00	0.385	1.00–1.00
**SGA**	202	0.82	0.37	0.656	0.34–1.99
**Writhing GMs**	199	1.56	0.30	0.019	1.07–2.27
**HNNE**	169	0.88	0.06	0.05	0.78–1.00

OR, odds ratio; Std. Err., Standard Error; CI, confidence interval; GA, gestational age; BW, birth weight; SGA, small for gestational age; GMs, general movements; HNNE, Hammersmith Neonatal Neurological Examination.

**Table 3 brainsci-12-00847-t003:** Multivariate logistic regression between early neonatal assessments/demographics and early diagnosis of CP/high-risk status at 3–4 months’ CA.

	N	OR	Std. Error	*p* Value	95% CI
**Sex**	169	0.87	0.43	0.776	0.33–2.31
**GA**	169	0.67	0.15	0.068	0.44–1.03
**BW**	169	1.00	0.00	0.619	1.00–1.00
**SGA**	169	1.42	1.04	0.633	0.34–5.94
**Writhing**	169	1.44	0.31	0.087	0.95–2.20
**HNNE**	169	0.91	0.06	0.142	0.79–1.03

OR, odds ratio; Std. Err., Standard Error; CI, confidence interval; GA, gestational age; BW, birth weight; SGA, small for gestational age; GMs, general movements; HNNE, Hammersmith Neonatal Neurological Examination.

**Table 4 brainsci-12-00847-t004:** Spearman’s rank correlation (r_s_) between early neonatal and infant assessments.

	Fidgety GMs					HINE				
	N	r_s_	Std. Err.	*p*-Value	95% CI	N	r_s_	Std. Err.	*p*-Value	95% CI
**Writhing GMs**	196	0.03	0.07	0.723	−0.12–0.17	191	−0.10	0.07	0.173	−0.24–0.04
**HNNE**	167	−0.18	0.07	0.012	−0.32–0.04	161	0.43	0.06	<0.001	0.31–0.56

GMs, general movements; HNNE, Hammersmith Neonatal Neurological Examination; HINE, Hammersmith Infant Neurological Exam; r_s_, correlation coefficient; Std. Err., Standard Error; CI, confidence interval.

**Table 5 brainsci-12-00847-t005:** Linear regression between HNNE/GA and HINE at 3–4 months’ CA.

	R	Std. Err.	*p*-Value	95% CI
**HNNE**	0.80	0.17	<0.001	0.47–1.13
**GA**	0.81	0.29	0.005	0.26–1.37

R, correlation coefficient; Std. Err., Standard Error; CI, confidence interval; HNNE, Hammersmith Neonatal Neurological Examination; HINE, Hammersmith Infant Neurological Examination; GA, gestational age.

## Data Availability

De-identified data available on reasonable request.

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
