# Peer review of "Assessing the Utility of Neonatal Screening Assessments in Early Diagnosis of Cerebral Palsy in Preterm Infantsâ€"

_brainsci, 2022, doi:10.3390/brainsci12070847_

Round 1
Reviewer 1 Report
Thank you for your work. I found this interesting to read and I can see this will be very insightful for a lot of researchers and clinicians working in this space.
Abstract
In the second last line of the results about Writhing, can you can you clarify if you are talking about abnormal or normal writhing (or both?).
Introduction
Perhaps to appease some critics (/those less comfortable with 'early diagnosis'), you may want to incorporate the phrasing early detection of infants at high risk of CP (or something along those lines) within the intro?
It seems like there is a bit of repetition in the introduction, for example 'Early intervention' is brought up 4 times. Have you considered flipping the order of introducing the HINE and HNNE? the GMs paragraph, the HINE paragraph and the sentence about combining for prediction could flow well, and then you could slot the HNNE further down when you talk about it in the third last paragraph.
Is it significant to point out that in some cases these infants may not be normally seen at 3-4 months within a usual clinic schedule (so a neonatal assessment may be more relevant to capture them?). I don't know if this is actually the case or not, it is just a thought.
Has there been any previous studies that have tracked the trajectory (or correlations) between HNNE and HINE at all? If not it might be worth noting that as it feels like a relevant gap.
Once you have abbreviated CP you have written it again in full a few time throughout the manuscript. Perhaps just double check this.
Methods
Within the clinic, I presume all infants have their writhing and HNNE as standard practice (i.e. all 393?)? Could you just state this to give this context? Also I presume that the clinic tries to get more than one writhing video, but that isn't clear. It was only brought to my attention when it came to the CS >2 group. Could some of the CS on one occasion group just be one occasion because they only had one video? Can you explain also why you chose to categorise the CS >2 group? My presumption is that you are talking more about if it is persistent across a trajectory of GMs assessments, but this hasn't been explained at all in your methods.
Also can you state what the specific diagnosis criteria was for CP at 3-4 mo (I presume Absent Fidgety + HINE <57 with/without neuroimaging findings?) and then also the criteria for the high risk? This is important to explain how/why the High Risk and the CP group were separated.
Who reviewed the videos (two people individually? blinding? were they already scored in practice or researchers after re-reviewing for the study?)
What age was the HNNE taken?
Results
Section 3.2: when you talk about abnormal writhing can you clarify that this is including PR + PR/CS + CS + CS>2? or are you taking out the PR group?
In addition to how you have presented the data in table 1, is there a way you could present the sort of trajectory or pathway of the combination of data?ie. of the CP group, how many had Normal writhing and normal FM, and an HNNE >27 and HINE >57?
I understand this could look complicated, but it would be interesting if you could present that (or part of that) in any way?
Figure 1. can you add the actual n in each bar? it might also be helpful to make the normal writhing bar a white bar with a black border
Figure 2. it could be interesting to marke the 27 cut off line on the graph for a visual comparison?
3.3. Is it worth noting again that 'normal' score was 0, so here is it that the HNNE was correlated with 'abnormal' writhing of varying levels of 'worseness'?
Figure 3. could it be interesting to add the cut off points for the HNNE and HINE, and then the participants with a CP or CP high risk be in triangles and the rest stay as dots?
Discussion
Again here, it would be helpful to clarify what you are including as abnormal writhing (and clarify that abnormal fidgety includes absent and abnormal?).
I'm not so confident such a strong conclusion from this data can be made (but perhaps if some information clarifies some of my queries it clearer for me?). The medians and ranges of the HNNE between the CP and non CP group are very similar (despite the p value). The number of PR is very high in both groups, and I had wondered if this just might be so common to expect it within this group of HR infants. What if we only viewed CS as the 'risk' for CP, we have 77% had normal or PR in the Non-CP group, compared to 58% in the CP group? that still doesn't bring much confidence for the writhing GMA interpretation. Would it be more insightful if we could know their combination GM and HNNE score? Or if you would delve into the infants that had a 'delay' diagnosis at 3-4 months...are they the infants with the low HNNE and abnormal GMs in the 'non CP group'?
Perhaps this 'worsening score of GMs' may need to be discussed when talking of the outcomes of the writhing GMs?
Author Response
Reviewer 1
Thank you for your work. I found this interesting to read and I can see this will be very insightful for a lot of researchers and clinicians working in this space.
Thank you so much for your insightful comments and constructive criticism.
Abstract
In the second last line of the results about Writhing, can you can you clarify if you are talking about abnormal or normal writhing (or both?).
Thank you. We have clarified – it is abnormal writhing movements.
Introduction
Perhaps to appease some critics (/those less comfortable with 'early diagnosis'), you may want to incorporate the phrasing early detection of infants at high risk of CP (or something along those lines) within the intro?
We have clarified this, and the edits for the comment below as well, we think this will help with the overall clarity of this point.
It seems like there is a bit of repetition in the introduction, for example 'Early intervention' is brought up 4 times. Have you considered flipping the order of introducing the HINE and HNNE? the GMs paragraph, the HINE paragraph and the sentence about combining for prediction could flow well, and then you could slot the HNNE further down when you talk about it in the third last paragraph.
Thank you – we have rearranged the paragraphs & edited to flow better.
Is it significant to point out that in some cases these infants may not be normally seen at 3-4 months within a usual clinic schedule (so a neonatal assessment may be more relevant to capture them?). I don't know if this is actually the case or not, it is just a thought.
Excellent point – most infants are followed up in clinic in our setting but there is always the possibility of non-attendance etc. Additionally, not all centres will necessarily have the same clinic structure. Thank you for pointing this out.
Has there been any previous studies that have tracked the trajectory (or correlations) between HNNE and HINE at all? If not it might be worth noting that as it feels like a relevant gap.
Searching the literature, we could not find any studies correlating these two assessments. Thank you for pointing this out, we have included this point in the paper.
Once you have abbreviated CP you have written it again in full a few time throughout the manuscript. Perhaps just double check this.
Thank you – checked and amended.
Methods
Within the clinic, I presume all infants have their writhing and HNNE as standard practice (i.e. all 393?)? Could you just state this to give this context? Also I presume that the clinic tries to get more than one writhing video, but that isn't clear. It was only brought to my attention when it came to the CS >2 group. Could some of the CS on one occasion group just be one occasion because they only had one video? Can you explain also why you chose to categorise the CS >2 group? My presumption is that you are talking more about if it is persistent across a trajectory of GMs assessments, but this hasn't been explained at all in your methods.
Thank you for raising these points – they are important distinctions. We have added in further detail to the methods to clarify this.
Also can you state what the specific diagnosis criteria was for CP at 3-4 mo (I presume Absent Fidgety + HINE <57 with/without neuroimaging findings?) and then also the criteria for the high risk? This is important to explain how/why the High Risk and the CP group were separated.
Diagnosis criteria were absent fidgety + suboptimal HINE and abnormal neuroimaging findings if available.
Early CP was diagnosed if these assessments were severely abnormal (e.g absent fidgety + very low HINE in context of severely abnormal imaging) and neurological features of early CP present – for e.g. spasticity, fisted adducted thumbs, widespread asymmetries
High risk of CP was diagnosed for infants those who met criteria – sometimes with mild or moderately low scores but were not definitively diagnosed as CP at 3-4 months due to lack of above features.
Who reviewed the videos (two people individually? blinding? were they already scored in practice or researchers after re-reviewing for the study?)
Clinicians scored these in practice (in patient, outpatient, clinic) – not re-reviewed for the study.
What age was the HNNE taken?
Between 35-45 weeks PMA (post menstrual age)
Results
Section 3.2: when you talk about abnormal writhing can you clarify that this is including PR + PR/CS + CS + CS>2? or are you taking out the PR group?
Thank you for raising this – it is all abnormal GMs (i.e. everything other than the Normal writhing group) as there was a positive correlation between writhing GM and outcome, and as we categorised writhing GMs on a scale of 0-4, the more abnormal the GMs, the higher the risk. We took the most abnormal GM (as many had multiple assessments).
In addition to how you have presented the data in table 1, is there a way you could present the sort of trajectory or pathway of the combination of data?ie. of the CP group, how many had Normal writhing and normal FM, and an HNNE >27 and HINE >57?
I understand this could look complicated, but it would be interesting if you could present that (or part of that) in any way?
Very interesting idea, thank you. We have attempted to do this however feel that the format required to present it is excessively complicated for the scope of this particular paper.
Figure 1. can you add the actual n in each bar? it might also be helpful to make the normal writhing bar a white bar with a black border
Thank you, we have made the suggested changes.
Figure 2. it could be interesting to marke the 27 cut off line on the graph for a visual comparison?
This is a great suggestion; but think that we would not want to highlight the 27 in a big way, as this may vary with different cohorts. HNNE and HINE depend on gestation assessed.
3.3. Is it worth noting again that 'normal' score was 0, so here is it that the HNNE was correlated with 'abnormal' writhing of varying levels of 'worseness'?
Noted – we have amended this section.
Figure 3. could it be interesting to add the cut off points for the HNNE and HINE, and then the participants with a CP or CP high risk be in triangles and the rest stay as dots?
Thank you for this suggestion – however as mentioned above, the cut-off scores vary with different cohorts, and thus these particular cut-offs are specific to this study.
Discussion
Again here, it would be helpful to clarify what you are including as abnormal writhing (and clarify that abnormal fidgety includes absent and abnormal?).
Thank you for pointing this out, we have incorporated this suggestion into the paper.
I'm not so confident such a strong conclusion from this data can be made (but perhaps if some information clarifies some of my queries it clearer for me?). The medians and ranges of the HNNE between the CP and non CP group are very similar (despite the p value). The number of PR is very high in both groups, and I had wondered if this just might be so common to expect it within this group of HR infants. What if we only viewed CS as the 'risk' for CP, we have 77% had normal or PR in the Non-CP group, compared to 58% in the CP group? that still doesn't bring much confidence for the writhing GMA interpretation. Would it be more insightful if we could know their combination GM and HNNE score? Or if you would delve into the infants that had a 'delay' diagnosis at 3-4 months...are they the infants with the low HNNE and abnormal GMs in the 'non CP group'?
Thank you for this comment & raising this issue. We have amended the conclusion to temper it slightly and account for the degree of significance in the univariate vs multivariate analyses. A combination of GM and HNNE score would likely be more helpful; we have noted this in the conclusion for future directions.
Perhaps this 'worsening score of GMs' may need to be discussed when talking of the outcomes of the writhing GMs?
Noted – we have added this in, thank you.

Reviewer 2 Report
This paper describes the relationship between assessments in the neonatal period and outcome of assessments performed at approximately 3-4 months of age with the intention of diagnosing CP.
The background gives reasonable information on the early diagnosis of cerebral palsy but should give a stronger rationale on the advantages of doing this on neonatal assessments rather than that obtained at 4 months. For instance, it is possible that some infants are transferred to secondary units before this time or that the Early Neurodevelopmental Clinic is a limited resource or that excessive travel is required.
Although semantics, I think it would be clearer to the reader if the words "correlate/correlation" were consistently used rather than "predict / prediction" regarding this relationship as it refers more to the two assessments than the final clinical outcome.
The methods are largely clear and informative. However, it would be helpful to the reader if more information was given on frequency and timing of the neonatal assessments and who performs these. Additionally, I note that at 3-4 months “diagnoses were made based on fidgety GMA and HINE in combination with available neuroimaging….”; however, no mention is made of neuroimaging in the neonatal assessments. Much of the cranial ultrasound data would be available at the time of neonatal assessments, particularly if these were performed between 36 weeks CGA and discharge. If the study objective was to assess ability to predict future CP then the study should include consideration of the neuroimaging. If the study objective is to correlate GMAs and HINE/HNNE then neither time point should include imaging. The introduction stresses “real-world” setting, which would imply use of all available data but either way this should be made clear in the methods.
The results are largely clear and informative. However, as per above, more details should be given on timing of neonatal assessments in the cohort. In table 1, it looks as though all 176 and 26 infants in each group have had Writhing GMA but the distribution of the 31 infants without HNNE is not clear as results given as median (IQR).
Also, as discussed above, it is a potential weakness that ultrasound findings are not included in the logistic regression model if the intention is to assess prediction of outcome rather than correlate the tests.
I am satisfied that the statistics are performed appropriately but the way they are presented could be improved (see below for examples).
Minor point: under secondary outcome the text describes the Spearman rho coefficient then states “…however, neither of these results were significant”. It would be easier for the reader if the non significant result was stated up front i.e. The writhing GMs had a non significant Spearman’s rho coefficient of ….
The Discussion is mostly helpful but the statement “Most notably, our study found a significant association between abnormal writhing GMs and CP/high-risk diagnosis ….When adjusted for demographic factors these factors were no longer significant” should be revised. The methods make it clear the intention was always to perform adjustment for demographic characteristics so the initial discussion would be better focused on the final result rather than an interim result even if that was significant.
This also applies to the conclusion, which states “support the utility”. Plus the conclusions part of the abstract, which states “were associated” before adjusted for demographic factors. Both of these should be based on the final results.
The Tables were clear and Figure 2 using box and whisker plot plus Figure 3 plotting regression were very informative.
The references were reasonable and current.
Author Response
Reviewer 2
This paper describes the relationship between assessments in the neonatal period and outcome of assessments performed at approximately 3-4 months of age with the intention of diagnosing CP.
Thank you for reviewing this paper & for your constructive feedback.
The background gives reasonable information on the early diagnosis of cerebral palsy but should give a stronger rationale on the advantages of doing this on neonatal assessments rather than that obtained at 4 months. For instance, it is possible that some infants are transferred to secondary units before this time or that the Early Neurodevelopmental Clinic is a limited resource or that excessive travel is required.
Thank you for pointing this out, we have clarified this.
Although semantics, I think it would be clearer to the reader if the words "correlate/correlation" were consistently used rather than "predict / prediction" regarding this relationship as it refers more to the two assessments than the final clinical outcome.
Amended, thank you.
The methods are largely clear and informative. However, it would be helpful to the reader if more information was given on frequency and timing of the neonatal assessments and who performs these. Additionally, I note that at 3-4 months “diagnoses were made based on fidgety GMA and HINE in combination with available neuroimaging….”; however, no mention is made of neuroimaging in the neonatal assessments. Much of the cranial ultrasound data would be available at the time of neonatal assessments, particularly if these were performed between 36 weeks CGA and discharge. If the study objective was to assess ability to predict future CP then the study should include consideration of the neuroimaging. If the study objective is to correlate GMAs and HINE/HNNE then neither time point should include imaging. The introduction stresses “real-world” setting, which would imply use of all available data but either way this should be made clear in the methods.
Thank you – we have edited to clarify details of the neonatal assessments.
As imaging was included in the 3-4 months matrix for assessment for diagnosis (as per guidelines), it cannot be then compared.
The results are largely clear and informative. However, as per above, more details should be given on timing of neonatal assessments in the cohort. In table 1, it looks as though all 176 and 26 infants in each group have had Writhing GMA but the distribution of the 31 infants without HNNE is not clear as results given as median (IQR).
Thank you for noting this – we have added N values in the table for both HNNE and HINE, which we hope clarifies the distribution of the missing values.
Also, as discussed above, it is a potential weakness that ultrasound findings are not included in the logistic regression model if the intention is to assess prediction of outcome rather than correlate the tests.
As noted above, ss imaging was included in the 3-4 months matrix for assessment for diagnosis (as per guidelines), it cannot be compared with neonatal assessments.
I am satisfied that the statistics are performed appropriately but the way they are presented could be improved (see below for examples).
Minor point: under secondary outcome the text describes the Spearman rho coefficient then states “…however, neither of these results were significant”. It would be easier for the reader if the non significant result was stated up front i.e. The writhing GMs had a non significant Spearman’s rho coefficient of ….
Thank you for this helpful suggestion, we have edited to suit.
The Discussion is mostly helpful but the statement “Most notably, our study found a significant association between abnormal writhing GMs and CP/high-risk diagnosis ….When adjusted for demographic factors these factors were no longer significant” should be revised. The methods make it clear the intention was always to perform adjustment for demographic characteristics so the initial discussion would be better focused on the final result rather than an interim result even if that was significant.
Agreed, thank you for highlighting this. we have amended this section.
This also applies to the conclusion, which states “support the utility”. Plus the conclusions part of the abstract, which states “were associated” before adjusted for demographic factors. Both of these should be based on the final results.
We have made appropriate edits to rectify these issues, thank you.
The Tables were clear and Figure 2 using box and whisker plot plus Figure 3 plotting regression were very informative.
Thank you very much for this feedback.
The references were reasonable and current.
Thank you for your comments & feedback!
